# Trait paranoia shapes inter-subject synchrony in brain activity during an ambiguous social narrative

Emily S. Finn [1], Philip R. Corlett[2], Gang Chen[3], Peter A. Bandettini[1] & R. Todd Constable [4]

Individuals often interpret the same event in different ways. How do personality traits modulate brain activity evoked by a complex stimulus? Here we report results from a naturalistic paradigm designed to draw out both neural and behavioral variation along a specific dimension of interest, namely paranoia. Participants listen to a narrative during functional MRI describing an ambiguous social scenario, written such that some individuals would find it highly suspicious, while others less so. Using inter-subject correlation analysis, we identify several brain areas that are differentially synchronized during listening between participants with high and low trait-level paranoia, including theory-of-mind regions. Follow-up analyses indicate that these regions are more active to mentalizing events in high-paranoia individuals. Analyzing participants' speech as they freely recall the narrative reveals semantic and syntactic features that also scale with paranoia. Results indicate that a personality trait can act as an intrinsic "prime," yielding different neural and behavioral responses to the same stimulus across individuals.

[1] Section on Functional Imaging Methods, Laboratory of Brain and Cognition, National Institute of Mental Health, Bethesda, MD 20892-9663, USA. [2] Department of Psychiatry, Yale School of Medicine, New Haven, CT 06511-6662, USA. [3] Scientific and Statistical Computing Core, National Institute of Mental Health, Bethesda, MD 20892-9663, USA. [4] Department of Radiology and Biomedical Imaging, Yale School of Medicine, New Haven, CT 06520-8042, USA. Correspondence and requests for materials should be addressed to E.S.F. (email: emily.finn@nih.gov)

That different individuals may see the same event in different ways is a truism of human nature. Examples are found at many scales, from low-level perceptual judgments to interpretations of complex, extended scenarios. This latter phenomenon is known as the "Rashomon effect"[1] after a 1950 Japanese film in which four eyewitnesses give contradictory accounts of a crime and its aftermath, raising the point that, for multifaceted, emotionally charged events, there may be no single version of the truth.

What accounts for these individual differences in interpretation? Assuming everyone has access to the same perceptual information, personality traits may bias different individuals toward one interpretation or another. Paranoia is one such trait, in that individuals with strong paranoid tendencies may be more likely to assign a nefarious interpretation to otherwise neutral events[2]. While paranoia in its extreme is a hallmark symptom of schizophrenia and other psychoses, trait-level paranoia exists as a continuum rather than a dichotomy[3,4]: on a behavioral level, up to 30% of people report experiencing certain types of paranoid thoughts (e.g., "I need to be on my guard against others") on a regular basis[5] and trait paranoia in the population follows an exponential, rather than bimodal, distribution[6].

Few neuroimaging studies have investigated paranoia as a continuum; the majority simply contrast healthy controls and patients suffering from clinical delusions. However, a handful of reports from subclinical populations describe patterns of brain activity that scale parametrically with tendency toward paranoid or delusional ideation. For example, it has been reported that higher-paranoia individuals show less activity in the medial temporal lobe during memory retrieval and less activity in the cerebellum during sentence completion[7], less activity in temporal regions during social reflection[8] and auditory oddball detection[9], but higher activity in the insula and medial prefrontal cortex (mPFC) during self-referential processing[10] and differential patterns of activity in these regions as well as the amygdala while viewing emotional pictures[11].

Such highly controlled paradigms enable precise inferences about evoked brain activity but potentially at the expense of real-world validity. For example, brain response to social threat is often assessed with decontextualized static photographs of unfamiliar faces presented rapidly in series[12]. Compare this to threat detection in the real world, which involves perceiving and interacting with both familiar and unfamiliar faces in a rich, dynamic social context. Paranoid thoughts that eventually reach clinical significance usually have a slow, insidious onset, involving complex interplay between a person's intrinsic tendencies and his or her experiences in the world. In studying paranoia and other trait-level individual differences, then, is important to complement highly controlled paradigms with more naturalistic stimuli.

Narrative is an attractive paradigm for several reasons. First, narrative is an ecologically valid way to study belief formation in action. Theories of fiction posit that readers model narratives in a Bayesian framework in much the same way as real-world information[13], and story comprehension and theory-of-mind processes share overlapping neural resources[14]. Second, a standardized narrative stimulus provides identical input, so any variation in interpretation reflects individuals' intrinsic biases in how they assign salience, learn, and form beliefs. Third, from a neuroimaging perspective, narrative listening is a continuous, engaging task that involves much of the brain[15] and yields data lending itself to innovative, data-driven analyses such as inter-subject correlation (ISC)[16,17].

Previous work has shown that experimenters can manipulate patterns of brain activity during naturalistic stimuli by explicitly instructing participants to focus on different aspects of the stimulus. For example, Cooper et al. reported that activity patterns in temporal and frontal regions varied according to whether listeners were told to pay attention to action-, space-, or time-related features of short stories[18]. Lahnakoski et al. showed participants the same movie twice, asking them to adopt different perspectives each time, and found differences in neural synchrony depending on which perspective had been taken[19]. Most recently, Yeshurun et al. presented participants with a highly ambiguous story with at least two plausible—but very different—interpretations and used explicit primes to bias each participant toward one interpretation or the other. Responses in higher-order brain areas, including default mode, were more similar among participants who had received the same prime, indicating that shared beliefs have a powerful effect on how individuals perceive an identical stimulus[20]. However, while informative, these studies have all relied on an explicit prime or instruction; they cannot explain why individuals often spontaneously arrive at different interpretations of the same stimulus.

In this work, we use participants' intrinsic personality traits as an implicit prime, relating individual differences in trait paranoia to brain activity during a naturalistic task in which participants are faced with complex, ambiguous social circumstances. Using an original narrative, we show that while much of the brain is synchronized across all participants during story listening, stratifying participants based on trait paranoia reveals an additional set of regions with stereotyped activity only among high-paranoia individuals; many of these are regions involved in theory of mind and mentalizing. An encoding model of the task suggests that these regions, including the temporal pole and mPFC, are particularly sensitive to "mentalizing events" when the main character is experiencing an ambiguous social interaction or explicitly reasoning about other characters' intentions. Finally, we measure participants' behavioral reactions to the narrative by analyzing their speech as they freely recall the story and identify semantic and syntactic features that vary dimensionally with trait paranoia. Together, results indicate that a personality trait, in this case paranoia, can modulate both neural and behavioral responses to a single stimulus across individuals.

## Results

**Behavioral data and task performance.** We created a fictional narrative to serve as the stimulus for this study. The narrative described a main character faced with a complex social scenario that was deliberately ambiguous with respect to the intentions of certain characters; it was designed such that different individuals would interpret the events as more nefarious and others as less so. A synopsis of the story is given in Supplementary Note 1.

Twenty two healthy participants listened to a pre-recorded audio version of the narrative (total duration = 21:50 min:s, divided into three parts) during functional magnetic resonance image (fMRI) scanning. Following each of the three parts, participants answered three challenging multiple-choice comprehension questions to ensure they had been paying attention. Performance was very accurate (15 of the 22 subjects answered 9/9 [100%] questions correctly, while 5 answered 8/9 [89%] correctly and 2 answered 7/9 [78%] correctly). Self-report data indicated that subjects generally found the narrative engaging and easy to pay attention to (engagement rating on a scale of 1 to 5: mean = 3.8, s.d. = 0.96, median = 4, median absolute deviation [m.a.d.] = 0.72; attention rating: mean = 4.1, s.d. = 0.87, median = 4, m.a.d. = 0.66).

During a separate behavioral visit 1 week prior to the scan, participants completed several self-report questionnaires and behavioral tasks to assess personality traits and cognitive abilities (see Fig. 1a for a schematic of the experimental protocol). Our primary measure of interest was subscale A from the Green et al. Paranoid Thoughts Scale[21] (GPTS-A), henceforth referred to as

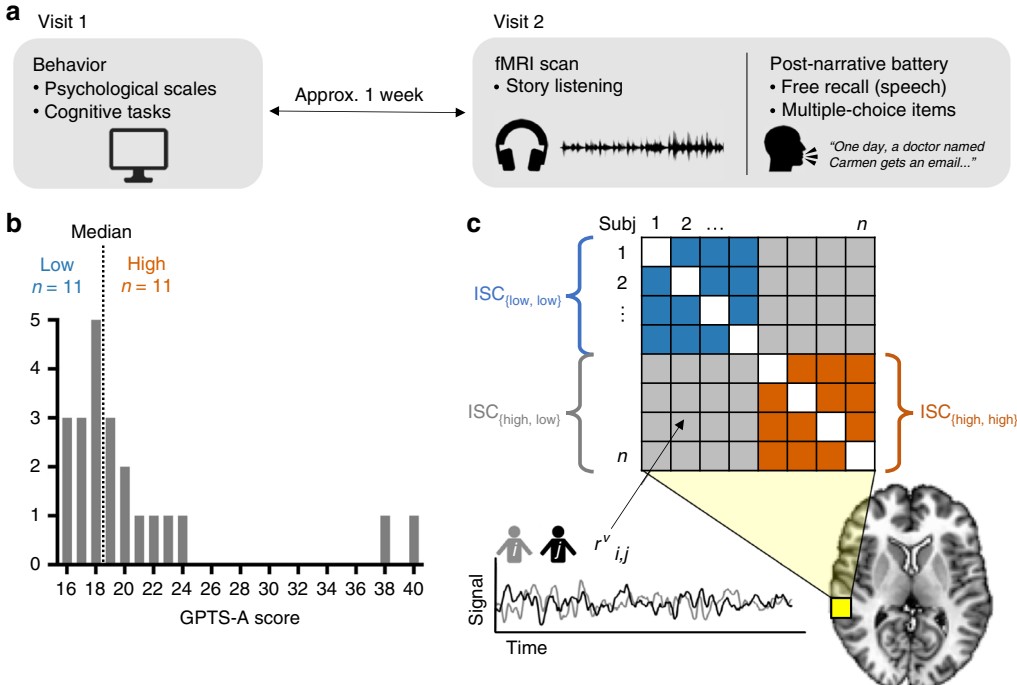

**Fig. 1** Experimental protocol, distribution of trait paranoia scores, and inter-subject correlation (ISC) analysis. **a** Schematic of the experimental protocol. Participants came to the laboratory for an initial behavioral visit, during which they completed several computerized cognitive tasks as well as self-report psychological scales, one of which was the Green et al. Paranoid Thoughts Scale (GPTS)[21]. To minimize demand characteristics and/or priming effects, the fMRI scan visit took place approximately 1 week later. During this visit, subjects listened to an ambiguous social narrative in the scanner and then completed an extensive post-narrative battery consisting of both free-speech prompts and multiple-choice items. **b** Distribution of scores on the GPTS-A subscale across $n = 22$ participants, and median split used to stratify participants into low ($\leq 18$, blue) and high ($\geq 19$, orange) trait paranoia. **c** Schematic of ISC analysis. Following normalization to a standard template, the ISC of activation time courses during narrative listening was computed for each voxel ($v$, yellow square; enlarged relative to true voxel size for visualization purposes) for each pair of subjects ($i,j$), resulting in a matrix of pairwise correlation coefficients ($r$ values). These values were then compared across paranoia groups using voxelwise linear mixed-effects models with crossed random effects to account for the non-independent structure of the correlation matrix[22]

trait paranoia score. We administered this scale on a different day and placed it among other tasks unrelated to paranoia to minimize any priming effects or demand characteristics that might influence participants' eventual reactions to the narrative. Possible scores on the GPTS-A range from 16 to 80; higher scores are generally observed only in clinical populations[21]. In our healthy sample, we observed a right-skewed distribution that nonetheless had some variance (range = 16–40, mean = 20.6, s.d. = 6.3; median = 18.5, m.a.d. = 4.0; see Fig. 1b for a histogram of the distribution). This is consistent with observations from much larger sample sizes that trait paranoia follows an exponential, rather than normal, distribution in the healthy population[5,6,21].

**Story listening evokes widespread neural synchrony**. Our primary approach for analyzing the fMRI data was ISC, which is a model-free way to identify brain regions responding reliably to a naturalistic stimulus across subjects[16,17]. In this approach, the time course from each voxel in one subject's brain across the duration of the stimulus is correlated with the time course of the same voxel in a second subject's brain. Voxels that show high correlations in their time courses across subjects are considered to have a stereotyped functional role in processing the stimulus. The advantage of this approach is that it does not require the investigator to have an a priori model of the task, nor to assume any fixed hemodynamic response function.

In a first-pass analysis, we calculated ISC at each voxel across the whole sample of $n = 22$ participants, using a recently developed statistical approach that relies on a linear mixed-

effects model with crossed random effects to appropriately account for the correlation structure of the data[22]. Results are shown in Fig. 2. As expected, given the audio-linguistic nature of the stimulus, ISC was highest in primary auditory cortex and language regions along the superior temporal lobe, but we also observed widespread ISC in other parts of association cortex, including frontal, parietal, midline, and temporal areas, as well as the posterior cerebellum. These results replicate previous reports that complex naturalistic stimuli induce stereotyped responses across participants in not only the relevant primary cortex but also higher-order brain regions[15,16,23].

Also as expected, ISC was generally lower or absent in primary motor and somatosensory cortex, although we did observe significant ISC in parts of primary visual cortex, despite the fact that there was no time course of visual input during the story. (To encourage engagement, we had participants fixate on a static photograph that was thematically relevant to the story during listening, so the observed ISC in visual cortex may reflect similarities in the time course of internally generated imagery across participants.)

**Paranoia modulates neural response to the narrative**. Having established that story listening evokes widespread neural synchrony across all participants, we next sought to determine whether there were brain regions whose degree of ISC was modulated by trait paranoia. Using a median split of GPTS-A scores, we stratified our sample into a low-paranoia group (GPTS-A ≤ 18, $n = 11$) and a high-paranoia group (GPTS-A ≥ 19,

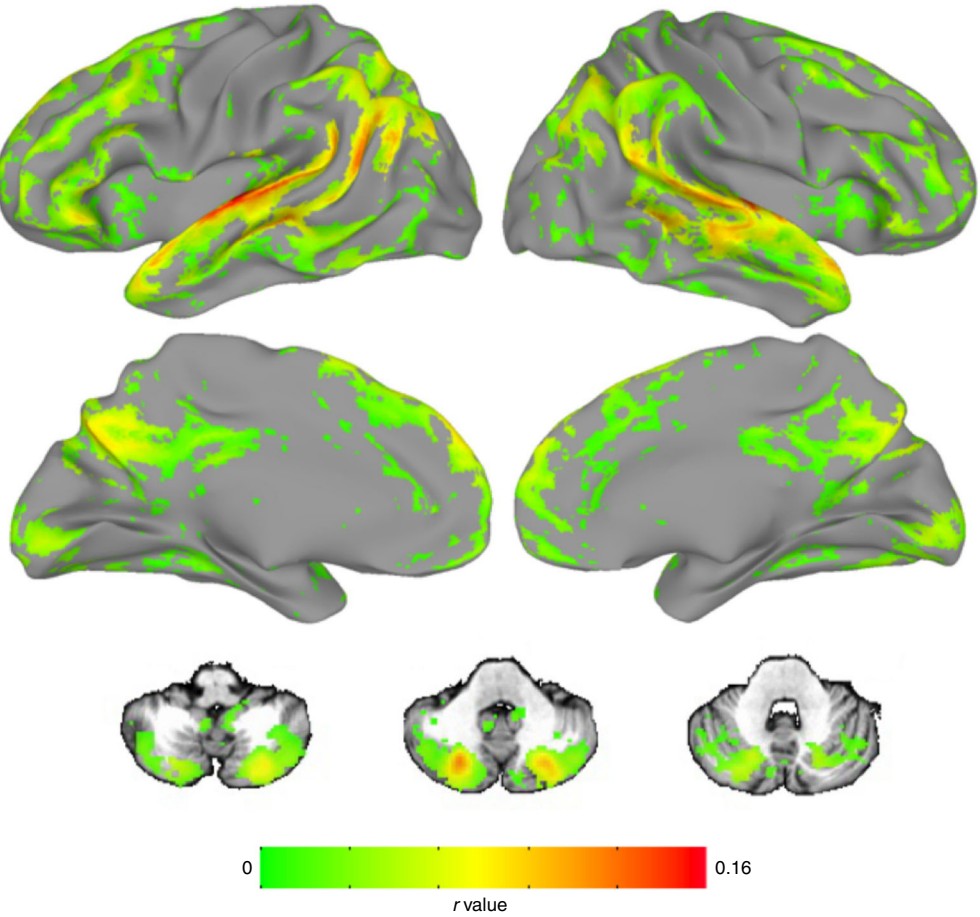

**Fig. 2** Narrative listening evokes widespread inter-subject correlation (ISC) across the whole sample. Voxels showing significant ISC across the time course of narrative listening in all participants ($n = 22$). As expected, the highest ISC values were observed in auditory cortex, but several regions of association cortex in the temporal, parietal, frontal, and cingulate lobes as well as the cerebellum also showed high synchrony. Also included are three representative axial slices from the cerebellum ($z$ coordinates in Talairach space: -38, -35, -29). Results are displayed at a voxelwise false-discovery rate (FDR) threshold of $q < 0.001$

$n = 11$) (Fig. 1b). We then used the same linear mixed-effects model described above formulated as a two-group contrast to reveal areas that are differentially synchronized across paranoia levels.

We opted for a median split rather than using raw paranoia score as a continuous covariate because of the unique challenge of an ISC-based analysis, which, to take advantage of all the information contained in the cross-subject correlation matrix (Fig. 1c), requires any covariates to be at the subject pair level, rather than the level of individual subjects. Because trait paranoia is a single scalar value per participant, it is difficult to calculate a meaningful pairwise metric. (Median splits can also mitigate the influence of extreme values, such as the two participants with GPTS-A ≥ 38 [cf. Fig. 1b], ensuring that these do not have an outsize effect on the results.) Still, we conducted post-hoc tests to investigate continuous relationships with raw GPTS-A score whenever possible to respect the inherently continuous nature of this trait and to facilitate interpretation.

We were primarily interested in three contrasts. First, which voxels show greater ISC among pairs of high-paranoia participants vs. low-paranoia participants, or vice versa? Second and third, which voxels show greater ISC among pairs of low- or high-paranoia participants, respectively (i.e., low–low or high–high), than pairs of participants mismatched for group (i.e., high–low)? All three contrasts reveals regions whose response time courses

are modulated by trait paranoia in some way. These contrasts are schematized in Fig. 1c.

Results are shown in Fig. 3. In the first contrast (Fig. 3a), several regions emerged as being more synchronized in the high-paranoia group relative to the low-paranoia group. Significant clusters were found in the left temporal pole (Talairach coordinates for center of mass: [+46.7, −10.0, −26.2]), left precuneus ([+10.8,+71.0,+35.9]), and two regions of the right mPFC (one anterior [−8.1, −46.9,+16.3] and one dorsal [+2.9, −14.8,+45.1]; Fig. 3a). Searches for these coordinates on Neurosynth, an automated fMRI result synthesizer for mapping between neural and cognitive states[24], indicated that, for the left temporal pole and right anterior mPFC clusters, top meta-analysis terms included "mentalizing," "mental states," "intentions," and "theory mind." There were no regions showing a statistically significant difference in the reverse direction (low paranoia > high paranoia).

In the second contrast (Fig. 3b, cool colors), pairs of low-paranoia participants were more synchronized than pairs of inter-group participants in the left lateral occipital gyrus (center of mass: [+31.3,+86.1,+14.0], Neurosynth: "objects," "scene," "encoding"), and in the third contrast (Fig. 3b, warm colors), pairs of high-paranoia participants were more synchronized than pairs of inter-group participants in the right angular gyrus ([−44.8,+57.9,+37.9], Neurosynth: "beliefs"). Interestingly, there were no voxels of statistically significant overlap between the

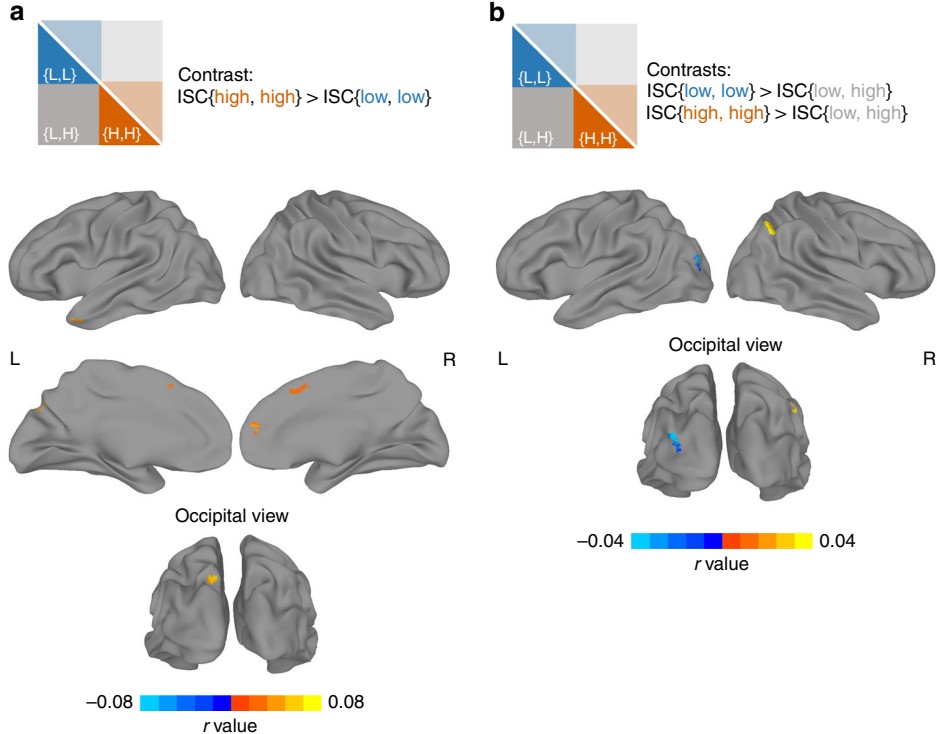

**Fig. 3** Trait paranoia modulates patterns of inter-subject correlation during narrative listening. **a** Results from a whole-brain, voxelwise contrast revealing brain regions that are more synchronized between pairs of high-paranoia participants than pairs of low-paranoia participants (contrast schematized in top panel, cf. Fig. 1c). Significant clusters were detected in the left temporal pole, two regions in the right medial prefrontal cortex (one anterior and one dorsal and posterior), and the left precuneus. No clusters were detected in the opposite direction (low > high). **b** Results from two whole-brain, voxelwise contrasts revealing brain regions that are more synchronized within a paranoia group than across paranoia groups. The first contrast (cool colors) revealed that left lateral occipital cortex was more synchronized within the low-paranoia group (i.e., low–low pairs) than across groups (i.e., high–low pairs; contrast schematized in top panel, cf. Fig. 1c). The second contrast (warm colors) revealed that right angular gyrus was more synchronized within the high-paranoia group (i.e., high–high pairs) than across groups. For all three contrasts, results are shown at an initial threshold of $p < 0.002$ with cluster correction corresponding to $p < 0.05$

second and third contrasts, indicating that no single region had a time course that was equally synchronized within groups but qualitatively different between groups. Instead, for most of the regions that emerged from the three contrasts, the relationship between trait paranoia and time course synchrony is best expressed by the Anna Karenina principle: all paranoid participants are alike; all not-paranoid participants are not paranoid in their own way (except in the lateral occipital gyrus, where it is the opposite).

As these regions were obtained via dichotomization into groups, we also conducted post-hoc tests to determine whether ISC remained sensitive to finer-grained differences in trait paranoia. We were primarily interested in two regions that emerged from the first contrast, the left temporal pole and right mPFC, since these are known from prior literature to be involved in theory of mind and mentalizing. To determine whether ISC in these regions scales monotonically with trait paranoia, we visualized the participant-by-participant ISC matrices with participants ordered by trait paranoia score (Fig. 4a, c). Visual inspection suggests a relatively continuous increase in ISC values as one moves down and to the right along the diagonal, which represents pairs of increasingly high-paranoia participants. To quantify this, we plotted each participant's median ISC with all other participants (i.e., the median of each row of the ISC matrix) against their paranoia rank within the sample (i.e., 1–22; Fig. 4b, d). For both regions of interest (ROIs), participants with higher

paranoia rank tended to have higher median ISC ($r_s = 0.71$ and $r_s = 0.63$ for the left temporal pole and right mPFC, respectively; both $p < 0.002$). We used paranoia rank rather than raw score to mitigate the influence of the two participants with extreme paranoia scores (≥38; cf. Fig. 1b).

**Effects are specific to paranoia.** We conducted several control analyses to rule out the possibility that the observed group differences were driven by a factor other than trait paranoia. (For all analyses in this section, we checked for both categorical and continuous relationships with paranoia; full results are reported in Table 1.)

For example, if the high-paranoia participants have better overall attentional and cognitive abilities, they might simply be paying closer attention to the story, inflating ISC values but not necessarily because of selective attention to ambiguous or suspicious details. However, there were no differences between high- and low-paranoia participants on any of the cognitive tasks we administered (verbal IQ, vocabulary, fluid intelligence or working memory), making it unlikely that observed differences are due to trait-level differences in attention or cognition. As for state-level attention during the story, there was no relationship between paranoia and number of comprehension questions answered correctly, total word count during the recall task, or self-report measures of engagement and attention. We also explored potential imaging-based confounds and found that

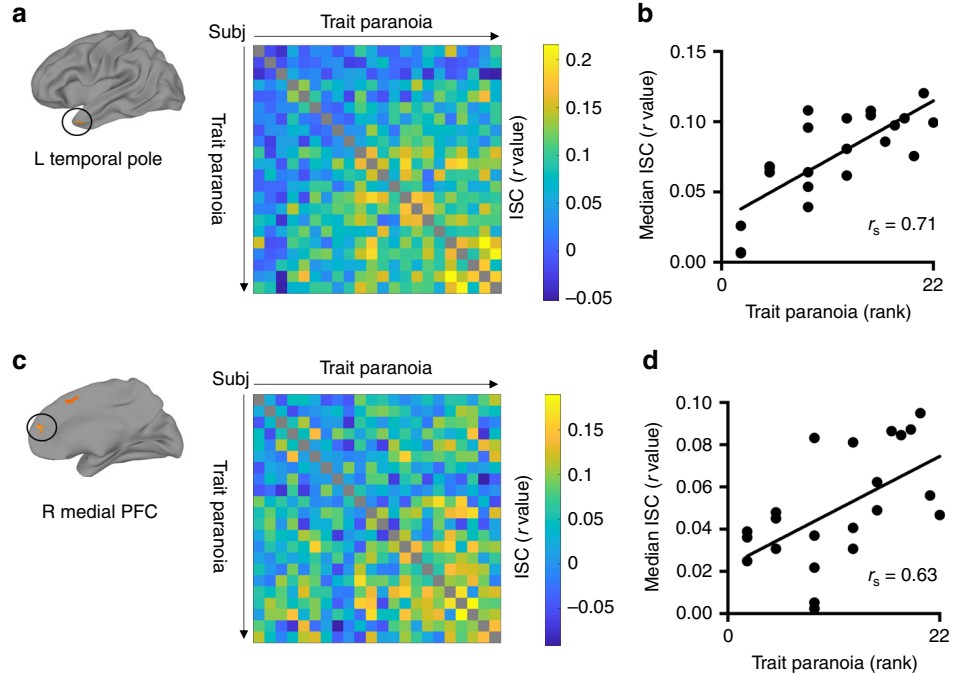

**Fig. 4** Inter-subject correlation (ISC) scales continuously with trait paranoia. Post-hoc analyses for two regions of interest (ROIs) that emerged from the dichotomized contrast between high- and low-paranoia groups (cf. Fig. 3a): left temporal pole (top row) and right medial prefrontal cortex (PFC, bottom row). **a** Location of ROI (left) and participant-by-participant ISC matrix (right) for the left temporal pole. Participants are ordered by increasing trait paranoia score. Each matrix element reflects the correlation between two participants' activation time courses in the left temporal pole during narrative listening. Higher correlations are visible as one moves to the right and down along the diagonal, representing pairs of increasingly high-paranoia individuals. **b** Scatter plot of paranoia rank vs. median ISC value—i.e., the median of each row of the ISC matrix in **a**. Each dot represents a participant. Rank correlation indicates a significant monotonic relationship between trait paranoia and median ISC in left temporal pole ($r_s = 0.71$, $p = 0.0002$). **c** Location of ROI and participant-by-participant ISC matrix for the right medial PFC. Participants are ordered as in **a**. **d** Scatter plot of each participant's paranoia rank vs. their median ISC value in the right medial PFC. As in **b**, rank correlation indicates a significant monotonic relationship between paranoia rank and median ISC ($r_s = 0.63$, $p = 0.0016$)

| Table 1 Trait paranoia was unrelated to potential confounding variables | | | | | |
|---|---|---|---|---|---|
| | | **Categorical (low vs. high)** | | **Continuous** | |
| | | *t* | *p* | **Spearman *r*** | *p* |
| Demographics | Age | 0.81 | 0.43 | −0.11 | 0.62 |
| | Sex[a] | 1.64 | 0.20 | — | — |
| | Education (years) | −0.24 | 0.81 | −0.15 | 0.49 |
| Cognitive ability | Working memory: Letter *n*-back (precision) | −0.45 | 0.66 | 0.16 | 0.47 |
| | Fluid intelligence: Raven's matrices (total correct of 9 items) | 0.00 | 1.00 | −0.03 | 0.89 |
| | Vocabulary: WRAT Word Reading (total correct of 42 items) | −1.42 | 0.17 | 0.31 | 0.16 |
| | Verbal IQ: Penn logical reasoning test (total correct of 8 items) | 0.23 | 0.82 | −0.01 | 0.96 |
| | Words of 6+ letters (free recall) | −1.03 | 0.32 | 0.04 | 0.85 |
| | Words per sentence (free recall) | 0.31 | 0.76 | −0.18 | 0.43 |
| fMRI data quality | Head motion (mean FD; mm) | 0.94 | 0.36 | 0.01 | 0.96 |
| | No. of frames censored | −0.70 | 0.49 | −0.08 | 0.74 |
| | Average tSNR | −1.12 | 0.28 | 0.23 | 0.30 |
| Attention to stimulus | No. of comprehension questions correct | −0.31 | 0.76 | 0.08 | 0.72 |
| | Total word count, free recall | 1.00 | 0.33 | −0.26 | 0.24 |
| | Self-reported attention | 0.48 | 0.63 | −0.02 | 0.95 |
| | Self-reported engagement | 0.89 | 0.39 | −0.10 | 0.65 |

There were no significant differences between high- and low-paranoia participants in terms of demographics, cognitive abilities, fMRI data quality, or attention to the stimulus. Categorical comparisons were carried out using Student's *t*-tests between the low- and high-paranoia groups as determined by median split (degrees of freedom for all *t*-tests = 20). Continuous comparisons were carried out using Spearman (rank) correlation between raw paranoia score and the variable of interest. All *p*-values are raw (uncorrected).
*FD* framewise displacement, *tSNR* temporal signal-to-noise ratio, *WRAT* Wide Range Achievement Test
[a]Measured with chi-squared test

paranoia was not related to amount of head motion during the scan (as measured by mean framewise displacement), number of censored frames, or temporal signal-to-noise ratio. Paranoia groups did not differ in age or sex breakdown. Thus we are reasonably confident that the observed effects are driven by true trait-level differences in paranoia between individuals.

**Activity to mentalizing events scales with paranoia.** Results of the first contrast from the two-group ISC analysis indicated that certain brain regions showed a more stereotyped response in high-paranoia vs. low-paranoia individuals. What features of the narrative were driving activity in these regions? In theory, ISC allows for reverse correlation, in which peaks of activation in a given region's time course are used to recover the stimulus events that evoked them[16]. In practice, this is often difficult. Especially with narrative stimuli, in which structure is built up over relatively long timescales[15], it is challenging to pinpoint exactly which event—word, phrase, sentence—triggered an increase in BOLD activity.

Rather than rely on reverse correlation, a data-driven decoding approach, we took an encoding approach: we modeled events in the task that we hypothesized would stimulate differing interpretations across individuals and evaluated the degree to which certain ROIs responded to such events, using a general linear model (GLM) analysis. Specifically, we labeled sentences in the story when the main character was experiencing an ambiguous (i.e., possibly suspicious) social interaction and/or sentences when she was explicitly reasoning about the intentions of other characters. For brevity, we refer to these time points as

"mentalizing events." In creating the regressor, all events were time-locked to the end of the last word of the labeled sentences, when participants are presumably evaluating information they just heard and integrating it into their situation model of the story.

We hypothesized that the two ROIs from the previous analysis known to be involved in theory of mind and mentalizing, the left temporal pole and right mPFC, would be more active to mentalizing events in individuals with higher trait paranoia. We included two additional ROIs, the left temporo-parietal junction (TPJ) and left Heschl's gyrus, as a positive and negative control, respectively. We selected the left TPJ as a positive control because of its well-established role in theory-of-mind and mentalizing processes and the fact that it emerged as highly synchronized across all participants (cf. Fig. 2) but did not show a group difference (cf. Fig. 3); thus we hypothesized that this region should respond to mentalizing events in all participants, regardless of trait paranoia. Conversely, left Heschl's gyrus (primary auditory cortex) should only respond to low-level acoustic properties of the stimulus and not show preferential activation to mentalizing events in either group or the sample as a whole. See Fig. 5a for ROI locations.

For each participant, we regressed the time course of each of these four ROIs against the mentalizing-events regressor and compared the resulting regression coefficients between groups (Fig. 5b). Compared to low-paranoia individuals, high-paranoia individuals showed stronger responses in both the left temporal pole (two-sample $t(20) = 2.71$, $p_{adj} = 0.014$) and right mPFC ($t(20) = 3.36$, $p_{adj} = 0.007$). As hypothesized, responses in the left TPJ were strong across the whole sample (one-sample $t(21) =$

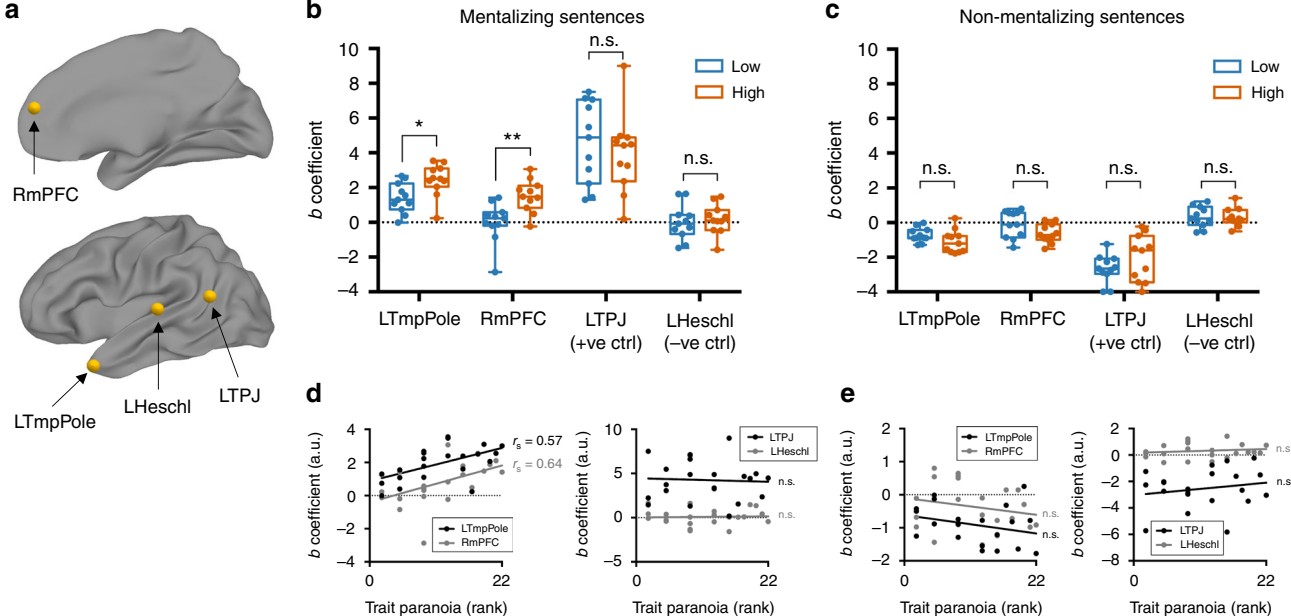

**Fig. 5** Response to mentalizing events is stronger in high- compared to low-paranoia individuals. **a** Regions of interest (ROIs) for the event-related analysis. LTmpPole, left temporal pole; RmPFC, right medial prefrontal cortex; LTPJ, left temporo-parietal junction; LHeschl, left Heschl's gyrus. **b** Comparison of beta coefficients for each ROI for the mentalizing-events regressor between the paranoia groups (low, blue; high, orange). Each dot represents a subject. Boxes represent the median and 25th/75th percentiles, and whiskers represent the minimum and maximum. *$p = 0.01$; **$p < 0.007$; n.s., not significant (p-values adjusted to control the false discovery rate at $Q = 5\%$). **c** Comparison of beta coefficients for each ROI for the non-mentalizing-events regressor (the inverse of the mentalizing-events regressor shown in **b**). Each dot represents a subject. Boxes represent the median and 25th/75th percentiles, and whiskers represent the minimum and maximum. **d** Post-hoc continuous analysis: Beta coefficients for the mentalizing-events regressor plotted against paranoia rank (coefficient values are the same as in **b**). Left panel: the two ROIs in which beta coefficient was hypothesized to scale with trait paranoia (LTmpPole and RmPFC). Right panel: the two control ROIs (LTPJ and LHeschl). Correlations between paranoia rank and beta coefficient: LTmpPole, $r_s = 0.57$, $p = 0.005$; RmPFC, $r_s = 0.64$, $p = 0.001$; LTPJ, $r_s = -0.04$, $p = 0.86$, LHeschl, $r_s = 0.02$, $p = 0.95$. **e** Beta coefficients for the non-mentalizing-events regressor plotted against paranoia rank (coefficients are the same as in **c**). Left and right panels as in **d**. Correlations between paranoia rank and beta coefficients (all n.s.): LTmpPole, $r_s = -0.28$, $p = 0.21$; RmPFC, $r_s = -0.22$, $p = 0.33$; LTPJ, $r_s = 0.085$, $p = 0.71$; LHeschl, $r_s = 0.17$, $p = 0.44$

8.73, $p < 0.0001$), but there was no significant difference between groups in this ROI ($t(20) = 0.67$, $p_{adj} = 0.34$). Also as hypothesized, the sample as a whole did not show a significant response to these events in primary auditory cortex (one-sample $t(21) = 0.44$, $p = 0.66$), and there was no group difference ($t(20) = 0.47$, $p_{adj} = 0.34$).

To confirm that these results hold if paranoia is treated as a continuous variable, we conducted additional post-hoc tests in which we correlated participants' paranoia ranks and regression coefficients for all four ROIs (Fig. 5d). As expected, response to suspicious events was significantly related to paranoia rank in the left temporal pole ($r_s = 0.57$, $p = 0.005$) and right mPFC ($r_s = 0.64$, $p = 0.001$) but not in the left TPJ ($r_s = -0.04$, $p = 0.86$) or left Heschl's gyrus ($r_s = 0.02$, $p = 0.95$).

As an additional control, to check that this effect was specific to mentalizing events and not just any sentence offset, we created an inverse regressor comprising all non-mentalizing events (i.e., by flipping the binary labels from the mentalizing-events regressor, such that all sentences were labeled except those containing an ambiguous social interaction or explicit mentalizing as described above). There were no differences between paranoia groups in any of the four ROIs in response to non-mentalizing sentences (Fig. 5c) and no continuous relationships between regression coefficient and paranoia rank (Fig. 5e). This indicates that trait paranoia is associated with differential sensitivity of the left temporal pole and right mPFC to not just any information but specifically to socially ambiguous information that presumably triggers theory-of-mind processes.

**Paranoia modulates behavioral response to the narrative.** Having established that trait paranoia modulates individuals' brain responses to an ambiguous narrative, we next sought to determine whether this trait also modulates their behavioral responses to the narrative. In other words, does trait-related (intrinsic) paranoia bear upon state-related (stimulus-evoked) paranoia? If the observed differences in neural activity propagate up to conscious perception and interpretation of the stimulus, then participants' subjective experiences of the narrative should also bear a signature of trait paranoia.

Immediately following the scan, participants completed a post-narrative battery that consisted of free-speech prompts followed by multiple-choice items to characterize their beliefs and feelings about the story. For the first item, participants were asked to retell the story in as much detail as they could remember, and their speech was recorded. Participants were allowed to speak for as long as they wished on whatever aspects of the story they chose. Without guidance from the experimenter, participants recalled the story in rich detail, speaking an average of 1081 words (range = 399–3185, s.d. = 610).

Audio recordings of participants' speech were transcribed and submitted to the language analysis software Linguistic Inquiry and Word Count[25] (LIWC). The output of LIWC is one vector per participant describing the percentage of speech falling into various semantic and syntactic categories. Example semantic categories are positive emotion ("love," "nice"), money ("cash," "owe"), and body ("hands," "face"), while syntactic categories correspond to parts of speech such as pronouns, adjectives, and prepositions; there are 67 categories in total.

Using partial least-squares regression, we searched for relationships between speech features and trait paranoia score. More than 72% of the variance in paranoia score could be accounted for by the first component of speech features; the loadings of semantic and syntactic categories for this component are visualized in Fig. 6a. The feature with the highest positive loading—indicating a positive relationship with paranoia—was affiliation, a category

of words describing social and familial relationships (e.g., "ally," "friend," "social"). Also associated with high trait paranoia was frequent use of adjectives as well as anxiety- and risk-related words (e.g., "bad," "crisis"); drives, a meta-category that includes words concerning affiliation, achievement, power, reward, and risk; and health-related words (e.g., "clinic," "fever," "infected"; recall that the story featured a doctor treating patients in a remote village; cf. Supplementary Note 1). Features with strongly negative loadings—indicating an inverse relationship with paranoia—included male references (e.g., "him," "his," "man," "father"); anger-related words ("yell," "annoyed"); function words ("it," "from," "so," "with"); and conjunctions ("and," "but," "until"). Figure 6b contains specific examples for selected categories from participants' speech transcripts.

After the free-speech prompts, participants answered a series of multiple-choice questions (see Supplementary Table 1 for the full questionnaire). First, they were asked to rate the degree to which they were experiencing various emotions (suspicion, paranoia, sadness, happiness, confusion, anxiety, etc; 16 in total) on a scale from 1 to 5. Most of ratings skewed low—for example, the highest paranoia rating was 3, and only six subjects rated their paranoia level >1. Interestingly, there was no significant correlation between trait paranoia score and self-reported paranoia ($r_s = -0.02$, $p = 0.91$) or suspicion ($r_s = 0.11$, $p = 0.62$) following the story. Neither were any of the other emotion ratings significantly correlated with trait-level paranoia (all uncorrected $p > 0.12$; see Fig. 6c).

Second, participants were asked to rate the three central characters on six personality dimensions (trustworthy, impulsive, considerate, intelligent, likeable, naive; see Supplementary Fig. 1a). Third, they were asked to rate the likelihood of each of the six scenarios (see Supplementary Fig. 1b), and finally, to indicate (via forced-choice options) what they believed the main character would do next, as well as what they themselves would do in her situation.

None of the individual questionnaire items significantly correlated with trait paranoia. However, to facilitate comparison with the speech data, we submitted the questionnaire data to a second partial least-squares regression to search for multi-dimensional relationships. This analysis revealed a first component of questionnaire responses that accounted for 62% of the variance in trait paranoia (Supplementary Fig. 1c). Features with the highest positive loadings, indicating a positive relationship with paranoia, included certain answers about what individuals thought the main character might do next as well as what they would do in her place (e.g., escape from the situation), as well as feeling more uncomfortable and suspicious following the story. Features with the highest negative loadings, indicating an inverse relationship with paranoia, included feeling more amused, inspired, and hopeful following the story, as well a tendency to agree with one of the scenarios ("Juan and the other villagers had not known anything about the disease before Carmen arrived").

Overall, then, we found signatures of paranoia in story-evoked behavior using both free-speech and self-report measures. Participants' free speech was slightly more sensitive than their answers on the multiple-choice questionnaire. Self-report is a coarse measure that may suffer from response bias; behavior provides a richer feature set that allows for the discovery of more subtle associations. In studying nuanced individual differences, then, these results highlight the desirability of capturing behavior in both traditional and naturalistic ways.

**Discussion**

Here we have shown that a personality trait can act as a lens, or "implicit prime," through which individuals perceive ambiguous

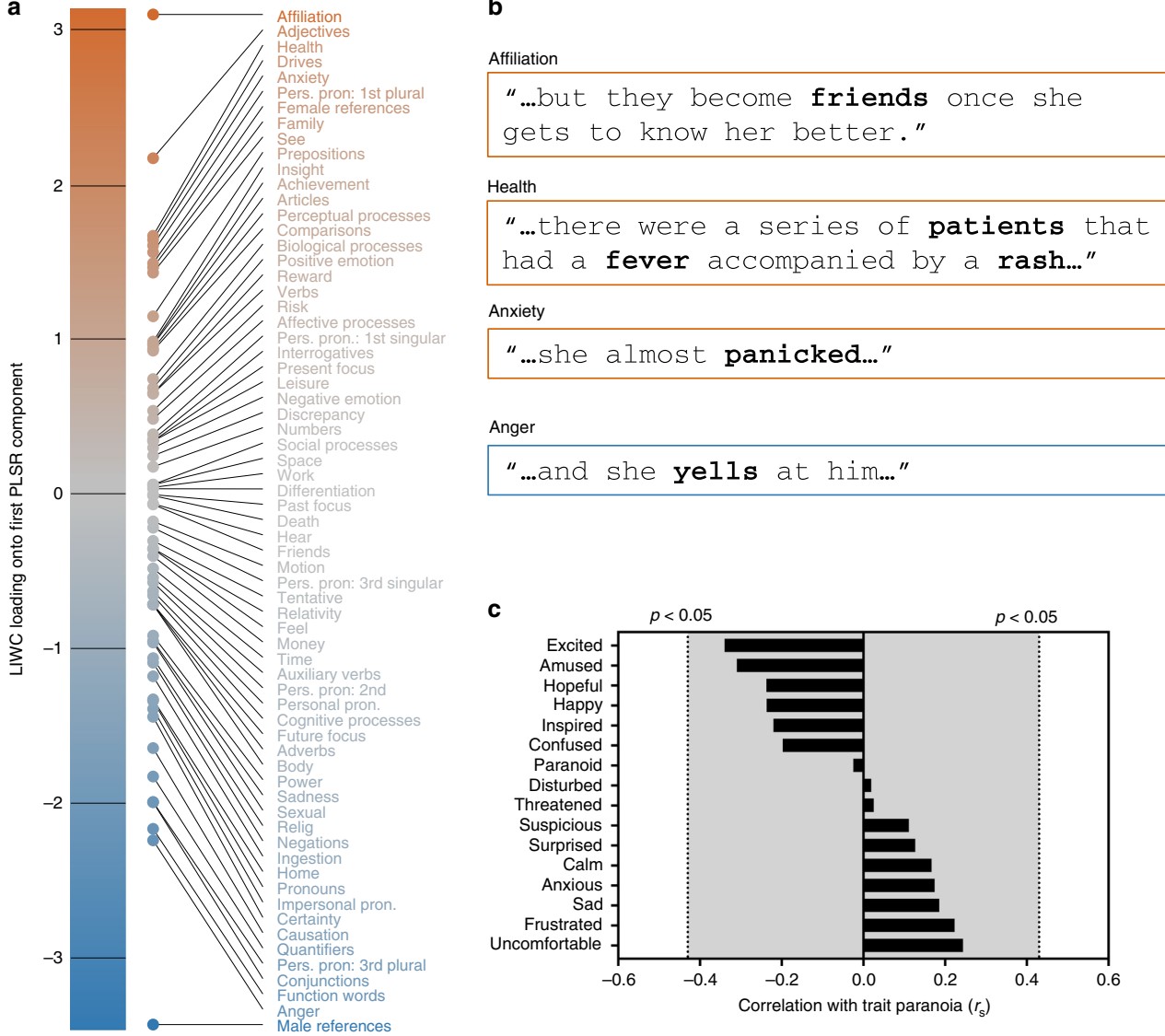

**Fig. 6** Speech analysis reveals a signature of trait paranoia in behavioral response to the narrative. **a** Loadings of all semantic and syntactic categories for the first component from a partial least squares regression relating features of speech during narrative recall to trait paranoia score, sorted by strength and direction of association with paranoia (those positively related to paranoia at top in orange; those inversely related at bottom in blue). **b** Example sentences from participant speech transcripts containing words falling into the three of the top positive categories (affiliation, health, and anxiety) and one of the top negative categories (anger). **c** Rank correlations between participants' trait-level paranoia and their self-report measures of 16 emotions following the narrative (self-report was based on a Likert scale from 1 to 5). Dotted lines represent approximate threshold for a significant correlation at $p < 0.05$ (uncorrected). Gray-shaded area indicates non-significance

events, shaping both their neural and behavioral responses to an identical stimulus. Previous work using naturalistic tasks has shown that brain activity and behavioral responses are sensitive to experimenter instructions, i.e., an explicit prime[19,20], or to the nature of the stimulus itself, i.e., whether it is more or less compelling or entertaining[26–28]. The present study extends these results in an important new direction, suggesting that there is substantial implicit variation in the brain's response to a naturalistic stimulus that stems from trait-level individual differences.

Our results have implications for the neural correlates of both trait- and state-related paranoia. Those with higher trait paranoia may have more stereotyped brain responses because suspicious and/or paranoid schemas come to mind more readily for these individuals; the idea that certain individuals tend to engage certain constructs more frequently across time and situations has been termed "chronic accessibility"[29]. The relative hyperactivity

of theory-of-mind regions to mentalizing events in high-paranoia individuals fits with the conception of paranoia as "over-mentalizing" or the tendency to excessively attribute (malevolent) intentions to other people's actions[30]. Both regions of differential response, the temporal pole and mPFC, are sometimes, but not always, reported in theory-of-mind tasks broadly construed[31]; individual differences may at least partially explain the inconsistencies in the literature.

While the present study included only healthy controls with subclinical paranoia, it may provide a useful starting point for the study of paranoid or persecutory delusions in schizophrenia and related illnesses. Delusions with a persecutory theme account for roughly 70–80% of all delusions. This high prevalence is stable across time[32] and geo-cultural factors[33–36], suggesting a strong biological component. Persecutory delusions are also the type most strongly associated with anger and most likely to be acted

upon, especially in a violent manner[37]. Thus understanding the neurobiological basis of paranoid delusions is a critical problem in psychiatry.

But because delusions typically have a slow, insidious onset, it is nearly impossible to retrospectively recover triggering events in individual patients. A related challenge is that, while thematically similar, each patient's delusion is unique in its details. Thus it is difficult to devise material that will evoke comparable responses across patients. One solution is to craft a model context using a stimulus that is ambiguous yet controlled—i.e., identical across participants, permitting meaningful comparisons of time-locked evoked activity—such as the one used in this work. Paradigms such as this one may shed light on mechanisms of delusion formation and/or provide eventual diagnostic or prognostic value.

While there is little work investigating brain activity during naturalistic stimuli in psychiatric populations, a handful of studies have used such paradigms in autism, finding that autistic individuals are less synchronized with one another and with typically developing controls while watching movies of social interactions[38–40]. Notably, the degree of asynchrony scales with autism-spectrum phenotype severity in both the patient and control groups[39]. It is interesting to juxtapose these reports with the present results, in which individuals with a stronger paranoia phenotype were more synchronized during exposure to socially relevant material; ultimately, this fits with the notion of autism and psychosis as opposite ends of the same spectrum, involving hypo- and hyper-mentalization, respectively[41,42]. Future studies should combine naturalistic stimuli with ISC-based analyses that cut across diagnostic labels to examine how neural responses vary across the full range of human phenotypes.

From a methodological perspective, much of the fMRI research on individual differences has shifted in recent years from measuring activation in task-based conditions to measuring functional connectivity, predominantly at rest[43–47]. Both paradigms suffer from limitations: traditional tasks are so tightly controlled that they often lack ecological validity; resting-state scans, on the other hand, are entirely unconstrained, making it difficult to separate signal from noise. Naturalistic tasks may be a happy medium for studying both group-level functional brain organization as well as individual differences[48,49]. We and others argue that such tasks could serve as a "stress test" to draw out individual variation in the brain and behaviors of interest[50–54], enhancing signal in the search for neuroimaging-based biomarkers and permitting more precise inferences about the sources of individual differences in neural activity.

## Methods

**Participants**. A total of 23 healthy volunteers participated in this study. Data from one participant were excluded owing to excessive head motion and self-reported falling asleep during the last third of the narrative. Thus, the final data set used for analysis contained 22 participants (11 females; age range = 19–35 years, mean = 27, s.d. = 4.4). All participants were right-handed, native speakers of English, with no history of neurological disease or injury, and were not on psychoactive medication at the time of scanning. All participants provided written informed consent in accordance with the Institutional Review Board of Yale University. The experiment took place over two visits to the laboratory. Participants were paid $25 upon completion of the first visit (behavioral assessments) and $75 upon completion of the second visit (MRI scan); all participants completed both visits.

**Stimulus**. An original narrative was written by author E.S.F. to serve as the stimulus for this experiment. For a synopsis of the story, see Supplementary Note 1. The full audio recording, as well as a complete transcript, are available in the "stimuli" directory at the following URL: https://openneuro.org/datasets/ds001338/. To mitigate confounds associated with education level or verbal IQ, we wrote the narrative text to be easy to comprehend, with a readability level of 78.1/100 and a grade 5.5 reading level as calculated by the Flesch–Kinkaid Formula.

A male native speaker of English read the story aloud and his speech was recorded using high-quality equipment at Haskins Laboratories (New Haven, CT).

The speaker was instructed to read in a natural, conversational tone, but without excess emotion. The final length of the audio recording was 21:50.

**Experimental protocol**. The experiment took place over two visits to the laboratory. Visit 1 was purely behavioral and took place approximately 1 week prior to visit 2 (MRI scan). During visit 1, participants completed a battery of self-report and behavioral tasks. While our primary measure of interest was the GPTS[21], we also administered several other psychological scales and cognitive assessments, in part to help reduce any demand characteristics that would allow participants to intuit the purpose of the study. We chose the GPTS because it provides a meaningful assessment of trait-level paranoia in clinical, but crucially, also in subclinical and healthy populations. In a previous study, score on this scale best predicted feelings of persecution following immersion in a virtual-reality environment[55]. The full GPTS contains two subscales, A and B, which pertain to ideas of social reference and ideas of persecution, respectively. We focused on subscale A, as it produces a wider range of scores in subclinical populations[21].

The following cognitive tests were administered via the web interface of the University of Pennsylvania Computerized Neuropsychological Test Battery (PennCNP; http://penncnp.med.upenn.edu)[56]: SRAVEN (short Raven's progressive matrices, a measure of abstraction and mental flexibility, or fluid intelligence); SPVRT (short Penn logical reasoning test, a measure of verbal intelligence); and LNB2 (letter n-back, a measure of working memory). We also administered the word reading test from the Wide Range Achievement Test 3[57], a measure of reading and vocabulary.

Visit 2 consisted of the MRI scan. The full audio recording was divided into three segments of length 8:46, 7:32, and 5:32, respectively; each of these segments was delivered in a continuous functional run while participants were in the scanner. To ensure attention, after each run, subjects answered three challenging multiple-choice comprehension questions regarding the content of the part they had just heard, for a total of nine questions. Immediately upon exiting the scanner, participants completed a post-narrative questionnaire that consisted of open-ended prompts to elicit free speech, followed by multiple-choice items. These are described further below.

**MRI data acquisition and preprocessing**. Scans were performed on a 3 T Siemens TimTrio system at the Yale Magnetic Resonance Research Center. After an initial localizing scan, a high-resolution three-dimensional volume was collected using a magnetization-prepared rapid gradient echo sequence (208 contiguous sagittal slices, slice thickness = 1 mm, matrix size 256 × 256, field of view = 256 mm, TR = 2400 ms, TE = 1.9 ms, flip angle = 8°). Functional images were acquired using a multiband T2*-sensitive gradient-recalled single-shot echo-planar imaging pulse sequence (TR = 1000 ms, TE = 30 ms, voxel size = 2.0 mm³, flip angle = 60°, bandwidth = 1976 Hz/pixel, matrix size = 110 × 110, field of view = 220 mm × 220 mm, multiband factor = 4).

We acquired the following functional scans: (1) an initial eyes-open resting-state run (6:00/360 TRs in duration) during which subjects were instructed to relax and think of nothing in particular; (2) a movie-watching run using Inscapes[58] (7:00/420 TRs); (3) three narrative-listening runs corresponding to parts 1, 2, and 3 of the story (21:50/1310 TRs in total); and (4) a post-narrative, eyes-open resting-state run (6:00/360 TRs) during which subjects were instructed to reflect on the story they had just heard. The present work focuses exclusively on data acquired during narrative listening. The narrative stimulus was delivered through MRI-compatible audio headphones and a short "volume check" scan was conducted just prior to the first narrative run to ensure that participants could adequately hear the stimulus above the scanner noise. To promote engagement, during the three narrative runs, participants were asked to fixate on a static image of a jungle settlement and to actively imagine the story events as they unfolded.

Following conversion of the original DICOM images to NIFTI format, AFNI (Cox 1996) was used to preprocess MRI data. The functional time series went through the following preprocessing steps: despiking, head motion correction, affine alignment with anatomy, nonlinear alignment to a Talairach template (TT_N27), and smoothing with an isotropic full-width half-maximum of 5 mm. A ventricle mask was defined on the template and intersected with the subject's cerebrospinal fluid mask to make a subject-specific ventricle mask. Regressors were created from the first three principal components of the ventricles, and fast ANATICOR (Jo et al. 2010) was implemented to provide local white matter regressors. Additionally, the subject's six motion time series, their derivatives, and linear polynomial baselines for each of the functional runs were included as regressors. Censoring of time points was performed whenever the per-time motion (Euclidean norm of the motion derivatives) was ≥0.3 or when ≥10% of the brain voxels were outliers. Censored time points were set to zero rather than removed altogether (this is the conventional way to do censoring, but especially important for ISC analyses, to preserve the temporal structure across participants). The final output of this preprocessing pipeline was a single functional run concatenating data from the three story runs (total duration = 21:50, 1310 TRs). All analyses were conducted in volume space and projected to the surface for visualization purposes.

We used mean framewise displacement (MFD), a per-participant summary metric, to assess the amount of head motion in the sample. MFD was overall relatively low (after censoring: mean = 0.075 mm, s.d. = 0.026, range = 0.035–0.14). Number of censored time points during the story was overall low but

followed a right-skewed distribution (range = 0–135, median = 4, median absolute deviation = 25). All 22 participants in the final analysis retained at least 89% of the total time points in the story, so missing data was not a substantial concern. Still, we performed additional control analyses to ensure that number of censored time points and amount of head motion were not associated with paranoia score in any way that would confound interpretation of the results (see Table 1).

**Inter-subject correlation**. Following preprocessing, ISC during the story was computed across all possible pairs of subjects $(i, j)$ using AFNI's 3dTcorrelate function, resulting in 231 ($n*(n-1)/2$, where $n = 22$) unique ISC maps, where the value at each voxel represents the Pearson's correlation between that voxel's time course in subject $i$ and its time course in subject $j$.

To identify voxels demonstrating statistically significant ISC across all 231 subject pairs, we performed inference at the single-group level using a recently developed linear mixed-effects (LME) model with a crossed random-effects formulation to accurately account for the correlation structure embedded in the ISC data[22]. This approach has been characterized extensively, including a comparison to non-parametric approaches, and found to demonstrate proper control for false positives and good power attainment[22]. The resulting map was corrected for multiple comparisons and thresholded for visualization using a voxelwise false discovery rate threshold of $q < 0.001$ (Fig. 2).

In a second analysis, we stratified participants according to a median split of scores on the GPTS-A subscale. We used these groups to identify voxels that had higher ISC values within one paranoia group or the other or higher ISC values within rather than across paranoia groups. To this end, we used a two-group formulation of the LME model. This model gives the following outputs: voxelwise population ISC values within group 1 ($G_{11}$); voxelwise population ISC values within group 2 ($G_{22}$); voxelwise population ISC values between the two groups that reflect the ISC effect between any pair of subjects with each belonging to different groups ($G_{12}$). These outputs can be compared to obtain several possible contrasts. Here we were primarily interested in three of these contrasts: (1) $G_{11}$ vs. $G_{22}$, (2) $G_{11}$ vs. $G_{12}$, and (3) $G_{22}$ vs. $G_{12}$. The maps resulting from each of these contrasts were thresholded using an initial voxelwise threshold of $p < 0.002$ and controlled for family-wise error (FWE) using a cluster size threshold of 50 voxels, corresponding to a corrected $p$-value of 0.05. We opted for a particularly stringent initial $p$-threshold in light of recent concerns about false positives arising from performing cluster correction on maps with more lenient initial thresholds[59].

**Event-related analysis**. A forced-aligner (Gentle; https://lowerquality.com/gentle/) was used to obtain precise timing information for each word in the narrative, by aligning the audio file with its transcript. One of the authors (E.S.F.) manually labeled sentences containing either an ambiguous social interaction or an instance of the main character mentalizing about other characters' intentions using a binary scoring system (1 = ambiguous social interaction or mentalizing present in sentence, 0 = neither ambiguous social interaction nor mentalizing present). Four additional, independent raters previously naive to the narrative listened to the same version that was played to participants in the scanner. They were then given a written version of the narrative broken down by sentence and asked to label each sentence as described above. Sentences that were labeled by at least three of the five raters were included in the final set of events. There were 48 sentences that met this criteria, with 17, 13, and 18 occurring in parts 1, 2, and 3 of the narrative, respectively.

Events were timestamped based on the TR corresponding to the offset of the last word of each labeled sentence. These timestamps were convolved with a canonical hemodynamic response function (HRF) to create the mentalizing-events regressor. Our assumption that evaluation and integration would happen primarily at the end of the sentence was based on theories of text comprehension, which hold that readers/listeners segment continuous linguistic information online into larger units of meaning or "macropropositions"; the mental models that listeners use to represent narratives are thus updated primarily at event boundaries[60–62]. Empirical neurobiological support for this comes from Whitney et al.[63], who showed, using a 23-min continuous narrative stimulus, that sentence boundaries coinciding with narrative shifts—defined as shifts in character, time, location, or action—evoked more brain activity than sentence boundaries not coincident with such shifts. Additional neuroimaging evidence comes from Zacks et al.[64], who demonstrated transient changes in brain activity that were time-locked to event boundaries during movie viewing.

However, some degree of evaluation and integration could also be happening online as participants listen to the event, and ideally the results from the regression would not depend on methodological choices about which parts of the sentence to model. To test this, we created a second version of the regressor, this time treating the entire sentence as a mini-block by modeling all TRs in each of the labeled sentences. Results were unchanged (see Supplementary Fig. 2). Thus we are confident that the results are robust to this methodological choice.

As a control analysis, we also created a regressor that was the inverse of the above regressor, by flipping the binary labels (0 or 1) for all sentences and convolving the corresponding sentences offset timestamps with the HRF; we refer to this as the non-mentalizing-events regressor.

ROIs for the GLM analysis were defined as follows. For the left temporal pole and right mPFC, ROIs were defined using the cluster-corrected group-comparison map for the contrast $ISC_{high} > ISC_{low}$ (cf. Fig. 3a). For the left TPJ and left Heschl's gyrus, spherical ROIs were created by placing a sphere with radius 4 mm around a central coordinate. In the case of the TPJ, this was the peak voxel in this region identified by the whole-sample ISC analysis (cf. Fig. 2; Talairach $xyz$, [+53, +55, +18]). In the case of Heschl's gyrus, this was selected anatomically (Talairach $xyz$, [−41, −24, +9]; as in Schönwiesner et al.[65]).

Time courses for each ROI were extracted from each participant's preprocessed functional data using AFNI's 3dmaskave function and regressed against both the mentalizing- and non-mentalizing-events regressors to obtain a regression coefficient for each participant for each ROI. These regression coefficients were then compared across groups using two-sample $t$-tests corrected for four multiple comparisons. In the case of the two control ROIs (TPJ and Heschl's gyrus) for the mentalizing-events regressor, these coefficients were also pooled across both groups and submitted to a one-sample $t$-test to test for a significant deviation from zero.

**Free-speech capture**. Immediately following their exit from the scanner, we gave participants the following prompts and recorded their speech: (1) "Please retell the story in as much detail as you can remember"; and (2) "What did you think of the story as a whole? In particular, did anything strike you as strange or confusing? How do you feel after listening to the story?" Here we focus on data acquired from the first prompt, as participants consistently talked for much longer to this one than to the second one (since they tended to preempt answers to second prompt in their answer to the first).

**Multiple-choice questionnaire**. Following the free-speech prompts, we had participants complete a computerized multiple-choice questionnaire to assess their feelings toward and beliefs about the story. A full list of items is provided in Supplementary Table 1; there were 47 in total.

**Analysis of speech features**. Audio recordings of participants' retelling of the story were professionally transcribed by a third-party company. We submitted the resulting transcripts to LIWC (www.liwc.net)[25], a software program that takes as input a given text and counts the percentage of words falling into different syntactic and semantic categories. Because LIWC was developed by researchers with interests in social, clinical, health, and cognitive psychology, the language categories were created to capture people's social and psychological states.

We restricted LIWC output to the 67 linguistic (syntactic and semantic) categories, excluding categories relating to metadata (e.g., percentage of words found in the LIWC dictionary), as well as categories irrelevant to spoken language (e.g., punctuation). Thus our final LIWC output was a $22 \times 67$ matrix where each row corresponds to a participant and each column to a category.

These categories can be scaled very differently from one another. For example, words in the syntactic category "pronoun" accounted for between 10.3 and 20.5% of speech transcripts, while words in the semantic category "leisure" accounted for only 0–1.09%. To give approximately equal weight to all categories, we standardized each category (to have zero mean unit variance) across participants before performing partial least squares regression (PLSR) as described in the next section. This ensures that the resulting PLS components are not simply dominated by variance in categories that are represented heavily in all human speech.

**Relating story-evoked behavior to paranoia**. To determine which speech features were most related to trait paranoia, we submitted the data to a PLSR with the $z$-scored speech features as $X$ (predictors) and trait paranoia score as $Y$ (response), implemented in Matlab as *plsregress*. PLSR is a latent variable approach to modeling the covariance structure between two matrices, which seeks to find the direction in $X$ space that explains the maximum variance in $Y$ space. It is well suited to the current problem, because it can handle a predictor matrix with more variables than observations, as well as multi-collinearity among the predictors.

In a first-pass analysis, we ran a model with 10 components to determine the number of components needed to explain most of the variance in trait paranoia. Results of this analysis indicated that the first component was sufficient to explain 72.3% of the total variance in paranoia score, so we selected just this component for visualization and interpretation. Feature loadings for this component are visualized in Fig. 6a.

In a parallel analysis, we submitted participants' answers to the multiple-choice questionnaire to a PLSR as the $X$ (predictor) matrix, again with paranoia score as the $Y$ (response) variable. Results of this analysis indicated that the first component was sufficient to explain 61.5% of the variance in paranoia score. Feature loadings for this component are visualized in Supplementary Fig. 1c.

**Code availability**. More information about this project, including links to code and other supporting material, can be found at: https://esfinn.github.io/projects/ParanoiaStory.html.

**Data availability**. Source data generated during this study, including raw MRI data and the full narrative stimulus (audio and text), are available at: https://openneuro.org/datasets/ds001338/.

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

## Acknowledgements

This research was supported by a National Science Foundation Graduate Research Fellowship to E.S.F. and by the Intramural Research Program of the NIMH (annual report ZIAMH002783). Portions of this study used the computational capabilities of the NIH HPC Biowulf cluster (hpc.nih.gov). The authors thank Paul Taylor for advice on pre-processing pipelines for naturalistic task data, Richard Betzel for sharing code to visualize component feature loadings (Fig. 6a and Supplementary Fig. 2), and Javier Gonzalez-Castillo, Daniel Handwerker, David Jangraw, Peter Molfese, Monica Rosenberg, and Tamara Vanderwal for helpful discussion.

## Author contributions

E.S.F., P.R.C., and R.T.C conceived the study. E.S.F. developed experimental materials and performed data collection. E.S.F. and G.C. analyzed the data. E.S.F., P.R.C., P.A.B., and R.T.C interpreted results. E.S.F. wrote the manuscript with comments from all other authors.

## Additional information

**Competing interests:** The authors declare no competing interests.

