## [Peer Review File · Nature Communications]

Reviewers' comments:

Reviewer #1 (Remarks to the Author):

The authors tested whether participants' brain response while listening to a narrative would be shaped by their trait-level paranoia. They found that participant's trait-level paranoia modulated the response in several regions, including theory of mind regions such as anterior temporal and medial prefrontal cortex. This is a very elegant study, that explores how the human brain process information in real-life experiences, with no external manipulations, and thus has high ecological validity. It is written in a very clear and stimulating manner. However, I do have some concerns regarding the analysis, that might undermine the authors' interpretation of the results.

Major concerns:

1. The main result of this study is that "there is substantial implicit variation in neural response to a naturalistic stimulus that stems from trait-level individual differences" (Pg 12).

In order to test for brain responses that are modulated by trait-level paranoia, the authors tested for voxels that show (i) greater ISC among pairs of high-paranoia participants versus low-paranoia participants, (Pg 6, ln 167-7), and voxels that show (ii) greater ISC among pairs of participants within the same paranoia group (i.e., high-high and low-low) than across groups (high-low) (Pg 6, ln 171-2).

I have questions regarding these two contrasts:

The first contrast revealed higher synchronization within high-paranoia participants in left temporal pole, left precuneus and right mPFC. However, it does not mean that these regions were more active in the high-paranoia group. For example, it could be that there was larger variability in the way low-paranoia participants interpreted the narrative, and that caused differences in the response in these regions (they were active, only with a different timecourse), which in turn resulted in lower ISC in this group. Thus, higher synchronization in the high-paranoia group does not imply that these regions were more active as suggested by the authors ("The relative hyperactivity of theory-of-mind regions in high-paranoia individuals" (Pg 12, ln 356).

The second contrast has the potential to test for paranoia-trait modulation of the brain response. This contrast revealed differences in the left middle occipital gyrus and left angular gyrus. However, it was not clear to me whether both within(high-paranoia) ISC and within(low-paranoia) ISC were larger than between(high-low paranoia) ISC? In order to avoid the same source of confound as in the first contrast, both within ISC needs to be significantly larger than the betweenISC.

As low ISC could result from different sources, I think that a more convincing way to show that trait-level paranoia shapes the neural response to the narrative is by classifying whether a specific participant has low or high paranoia trait based on the BOLD response in these regions.

2. The division of the participants into high and low trait-paranoia groups seems artificial. For example, five (out of 11) participants in the low group had paranoia score of 18, and 5 (out of 11) participants in the high group had paranoia score of 19-20. I wonder if there is a way to divide the participants into more meaningful (in terms of paranoia trait) groups based on their behavior – e.g. how they interpreted the described situations. This leads me to my next concern –

3. Was there a difference in the way the high- and low- paranoia participants interpreted the ambiguous parts of the story (the time intervals in which the "mentalizing" regressor =1)? I could not figure it out from the results' section. If not, how do the authors explain the differences in the regression coefficients (β) between the two paranoia groups?

Minor concerns:

4. The authors state that "In creating the regressor, all events were time-locked to the end of the last word of the labeled sentence when participants are presumably evaluating information they just heard and integrating it into their situation model of the story." (Pg 8, ln 230). Does this mean that the onset of an event ("1" value) of the regressor was the last word in the sentence? For how many TRs was it "1"? On which data do the authors base their assumption that evaluation and integration will happen only at the end of the sentence, and not online while participants listen to the event? This assumption has a major effect on the generation of the regressor and the resulted coefficients.

5. Why was the audio recording delivered in three consecutive runs, and not as a continuous narrative (Pg 24, ln 555)? Did one of the runs included parts of the narrative that were more ambiguous?

6. The max ISC seems to be $r=0.16$ (Fig 2), which is relatively low. What was the threshold of significant correlation ($r=?$)?

7. It is very difficult to read the labels in Figure 5a.

Reviewer #2 (Remarks to the Author):

This is a very interesting report showing that neural and behavioral responses to the same stimulus vary as a function of individual differences in trait paranoia.

The experimental paradigm used in this study is novel and ecologically valid.

The data collection and processing methods are well described and appear appropriate.

My only questions/suggestions to these authors are:

1) How does your sample means etc relate to the norms for the Paranoid Thoughts Scale?

2) Please justify your study design, i.e. creation of two groups by median split.

Reviewer #3 (Remarks to the Author):

This is quite rare for me, but I don't have a lot to say. I think this is a great paper. I found this work to be a compelling demonstration of the way that those high in paranoia process ambiguous social information differently than those low in paranoia and that this difference, at least in part, consists in greater mentalizing activity in the moments of ambiguity. I think more generally it adds to the literatures showing that high ISCs between individuals reflects more similar experience and processing of information as well as that various factors can prime us to experience things one way or another. To my knowledge this is the first study to use a personality variable as a chronic accessibility prime. The authors should probably look up Tory Higgins work on chronic accessibility since he made a similar argument decades ago (but without neuroimaging). Otherwise, though, I would strongly endorse this manuscript for publication.

We thank all three reviewers, as well as the editor, for their time and effort in considering our
work for publication. In what follows, we offer point-by-point responses to each of the reviewers'
concerns.

Both Reviewer 1 and Reviewer 2 raised concerns about the creation of high and low paranoia
groups via median split of scores on the Green et al. Paranoid Thoughts Scale A (GPTSA). We
agree with the reviewers, as well as much statistical literature, that in general it is preferable to
preserve the continuous nature of a variable whenever possible, rather than to dichotomize. In
fact, we explored several potential approaches that would allow us to use the raw GPTSA scores
as a covariate in an ISC analysis. However, the unique challenge of an ISC-based analysis is that
any covariates must be at the subject *pair* level, rather than the subject level. It is difficult to
calculate a meaningful trait metric at this level (i.e., should it be the sum of both participants'
paranoia scores? The difference? The ratio?). Thus, we decided that median split, while coarser,
was a more straightforward and ultimately interpretable test of our hypothesis.

We also note that in the simplest case of a single independent variable, as is the case here, a
median split is more conservative; that is, false negatives are possible, but false positives
unlikely, as power is reduced (Cohen, 1983). The fact that we still see significant differences
based on paranoia level indicates that this level of analysis is sufficient to detect effects. Further,
in the case of a non-normally distributed independent variable such as this one, median split has
another advantage, in that it reduces the influence of extreme values (here, the two subjects
scoring ≥ 38 , visible on the right side of the distribution in Fig. 1b.)

We have added this explanation and justification of the median-split approach to the revised
manuscript (paragraph beginning on line 164).

Despite the fact that our initial contrast was formulated as a dichotomy, we also checked for
continuous relationships with paranoia whenever possible. In the original manuscript, we
demonstrated that activation to mentalizing events for the two ROIs of interest (left temporal
pole and right medial PFC) scaled monotonically with paranoia using rank correlation between
beta weights and raw GPTSA scores (formerly Fig. 4, now Fig. 5d); this was above and beyond
the high vs. low group difference determined via t-test (now Fig. 5b). In the revised version, we
have also added an analysis showing that participants' median ISC in both left temporal pole and
right medial PFC relates to their trait paranoia in a continuous manner (Fig. 4). Thus, although
these ROIs were initially defined via a dichotomized contrast (median split), post-hoc analyses
indicate that there are continuous relationships—if not linear, at least monotonic—with raw GPTSA
score as well.

We also note some minor changes in this revised version of the manuscript not detailed in the
point-by-point responses below:

- 1. We have replaced most instances of 'trait-level paranoia' with 'trait paranoia' for brevity.
- 2. We have changed references to 'left posterior superior temporal sulcus' to 'left temporo-
parietal junction (TPJ)' to distinguish it from language regions along the superior temporal
lobe that were also highly synchronized, and to be more consistent with the theory-of-
mind literature.

- 3. We have added a partial least-squares regression between the multiple-choice
questionnaire data and trait paranoia, in order to provide a more fair comparison between
the questionnaire and free-speech data for capturing relationships between story-evoked
behavior and trait paranoia.
4. We now report the full details of the post-narrative multiple-choice questionnaire in
Supplementary Table S1, and its relationship to trait paranoia in Supplementary Fig. S1.

Reviewers' comments:

Reviewer #1 (Remarks to the Author):

The authors tested whether participants' brain response while listening to a narrative would be
shaped by their trait-level paranoia. They found that participant's trait-level paranoia modulated
the response in several regions, including theory of mind regions such as anterior temporal and
medial prefrontal cortex. This is a very elegant study, that explores how the human brain process
information in real-life experiences, with no external manipulations, and thus has high ecological
validity. It is written in a very clear and stimulating manner.

We thank the reviewer for his or her positive assessment of our work.

However, I do have some concerns regarding the analysis, that might undermine the authors'
interpretation of the results.

Major concerns:

1. The main result of this study is that "there is substantial implicit variation in neural response to a
naturalistic stimulus that stems from trait-level individual differences" (Pg 12).
In order to test for brain responses that are modulated by trait-level paranoia, the authors tested
for voxels that show (i) greater ISC among pairs of high-paranoia participants versus low-
paranoia participants, (Pg 6, ln 167-7), and voxels that show (ii) greater ISC among pairs of
participants within the same paranoia group (i.e., high-high and low-low) than across groups
(high-low) (Pg 6, ln 171-2).

I have questions regarding these two contrasts:

The first contrast revealed higher synchronization within high-paranoia participants in left
temporal pole, left precuneus and right mPFC. However, it does not mean that these regions were
more active in the high-paranoia group. For example, it could be that there was larger variability
in the way low-paranoia participants interpreted the narrative, and that caused differences in the
response in these regions (they were active, only with a different timecourse), which in turn
resulted in lower ISC in this group. Thus, higher synchronization in the high-paranoia group does
not imply that these regions were more active as suggested by the authors ("The relative
hyperactivity of theory-of-mind regions in high-paranoia individuals" (Pg 12, ln 356).

The reviewer is correct that if a region is less synchronized across participants, it does not
necessarily mean that the region is less active, or less involved in processing the stimulus, in any

individual participant: it could be that each low-paranoia participant simply has her or her own
idiosyncratic timecourse in the region. (To test this directly, we would need repeated
presentations of the narrative to evaluate within-participant consistency of activity patterns.)
However, it does mean that low-paranoia participants show markedly different timecourses than
high-paranoia participants, and from one another. This is perhaps most succinctly expressed by
the Anna Karenina principle: All paranoid people are alike; all un-paranoid people are un-
paranoid in their own way. Thus, trait-level paranoia does modulate the timecourse of activity in
these regions, just not in a “linear” way.

Still, the reviewer’s point is well taken, and in the revised manuscript we have taken care to
clarify our language, e.g. replacing statements that these regions are “more active” with statements
that they show “more stereotyped responses” in high- relative to low-paranoia participants.

While we cannot necessarily conclude that these ROIs are more active during the narrative
overall in high-paranoia participants, we can conclude from the post-hoc event-related analyses
that they are more active (i.e., show larger beta coefficients) specifically to mentalizing events.
We have clarified the Discussion to better reflect this point.

The second contrast has the potential to test for paranoia-trait modulation of the brain response.
This contrast revealed differences in the left middle occipital gyrus and left angular gyrus.
However, it was not clear to me whether both within(high-paranoia) ISC and within(low-
paranoia) ISC were larger than between(high-low paranoia) ISC? In order to avoid the same
source of confound as in the first contrast, both within ISC needs to be significantly larger than
the betweenISC.

The reviewer raises a good point. In the previous version of the manuscript, we had pooled all of
the within-group ISC values (high-high and low-low) together before comparing these to the
across-group values (high-low). However, any results from this contrast could be driven largely
by one group or the other; a significant result does not necessarily mean that each within- vs
across-group contrast would be significant on its own (high-high vs high-low, and low-low vs
high-low). For this we would need to compute each contrast separately, examine each output,
and then take the intersection.

Conducting the analysis in this way indicated that the within- > across-group effects in the two
regions previously identified, left lateral occipital and left angular gyrus, were driven largely by
the low and high groups, respectively. While for both regions, ISC values were numerically
higher for the intermediate group compared to across groups (e.g., for left lateral occipital, low-
low > high-high > high-low), the difference for the intermediate group (e.g., high-high > high-
low) was not statistically significant. We have therefore updated Fig. 3 to display the results of
each contrast separately.

This is not a confound per se, although it does change the interpretation slightly. All of these
contrasts—(1) high-high vs. low-low, (2) high-high vs. high-low and (3) low-low vs. high-low—test
for and reveal trait paranoia modulation of the brain response. There is simply no region of
overlap between contrasts. So, for example, we can conclude that if you have high trait paranoia,

you will show a stereotyped response (relative to other high scorers) in left temporal pole, and if
you have low trait-level paranoia you will *not* show a stereotyped response (relative to either
high or fellow low scorers) in that region, but you will show a stereotyped response in left
middle occipital gyrus.

As low ISC could result from different sources, I think that a more convincing way to show that
trait-level paranoia shapes the neural response to the narrative is by classifying whether a
specific participant has low or high paranoia trait based on the BOLD response in these regions.

In theory, we agree with the reviewer that performing out-of-sample prediction/classification
based on BOLD timecourses would be the logical next step. However, it would be circular to
limit the classifier to timecourses from these regions, which have been identified by analyzing
the whole group. We note that this was a relatively small, targeted study designed to serve as a
proof of concept that we can detect the influence of a personality trait on individuals' response to
a naturalistic stimulus using classical hypothesis testing methods. We believe these results
provide a foundation for future studies using larger data sets, which are better suited to the
challenge of out-of-sample prediction/classification.

2. The division of the participants into high and low trait-paranoia groups seems artificial. For
example, five (out of 11) participants in the low group had paranoia score of 18, and 5 (out of
11) participants in the high group had paranoia score of 19-20. I wonder if there is a way to
divide the participants into more meaningful (in terms of paranoia trait) groups based on their
behavior—e.g. how they interpreted the described situations. This leads me to my next concern—

While dividing participants based on their behavioral interpretations of the story might yield
interesting results in its own right, it would be a different question than the one we were trying to
answer here, since in that case we would be stratifying by *state-*, rather than trait-level, paranoia—
that is, transient paranoia induced by the stimulus. In the current study, we were interested in
how intrinsic or “baseline” paranoia modulates both brain response and behavioral response to the
story. For this, we need a trait-level scale that is independent of the story itself (i.e., the Green et
al. Paranoid Thoughts Scale).

We hope that the explanation at the beginning of this response letter, in which we address
concerns about the median split, as well as the continuous relationships that we now report
between brain activity and trait paranoia score for both the ISC (Fig. 4) and event-related results
(Fig. 5), are sufficient to convince the reviewer that this measure is a meaningful way to divide
participants.

3. Was there a difference in the way the high- and low- paranoia participants interpreted the
ambiguous parts of the story (the time intervals in which the “mentalizing” regressor =1)? I could
not figure it out from the results' section. If not, how do the authors explain the differences in the
regression coefficients (β) between the two paranoia groups?

We did not specifically probe how participants interpreted the ambiguous parts of the story (that
is, the mentalizing events modeled in the regressor), because we were concerned that asking
participants about specific events (either online or post-hoc) would distort their internally
generated reactions to the story—akin to “leading the witness”—and/or lead them to intuit the purpose
of the study. Thus, we opted to measure interpretation via a free-speech prompt (“Please retell the
story in as much detail as you can remember”) that was designed to be very neutral and allow
participants to speak about whatever parts of the story they chose. The trade-off of this decision
is that we can draw only general conclusions based on how participants spoke about the narrative
as a whole, as a proxy for their interpretation of the mentalizing content.

From the differences in regression coefficients, we conclude that trait paranoia confers increased
sensitivity to mentalizing events during online listening. Combining this with the behavioral
(speech) analysis, we conclude that trait paranoia modulates both brain response and behavioral
response to the story (paranoia → brain and paranoia → behavior). Still, we cannot necessarily
draw strong conclusions about how brain activity to individual events relates to ultimate
behavioral response (brain → behavior). We believe the Discussion section respects these
limitations on interpretation.

Minor concerns:

4. The authors state that “In creating the regressor, all events were time-locked to the end of the
last word of the labeled sentence when participants are presumably evaluating information they
just heard and integrating it into their situation model of the story.” (Pg 8, ln 230). Does this mean
that the onset of an event (“T” value) of the regressor was the last word in the sentence? For how
many TRs was it “T”? On which data do the authors base their assumption that evaluation and
integration will happen only at the end of the sentence, and not online while participants listen to
the event? This assumption has a major effect on the generation of the regressor and the resulted
coefficients.

Yes, the reviewer is correct that the onset of an event (“T” value) corresponded to the offset of the
last word of the labeled sentences. The “T” value lasted for one TR only. Our assumption that
evaluation and integration would happen primarily at the end of the sentence was based on
theories of narrative comprehension, which hold that readers/listeners segment continuous
linguistic information online into larger units of meaning, or “macropropositions”; the mental
models that listeners use to represent narratives are thus updated primarily at event boundaries
(Johnson-Laird, 1983; Van Dijk, Kintsch, & Van Dijk, 1983; Zwaan & Radvansky, 1998).
Empirical neurobiological support for this comes from (Whitney et al., 2009), who showed,
using a 23-minute continuous narrative stimulus, that sentence boundaries coinciding with
narrative shifts—defined as shifts in character, time, location, or action—evoked more brain activity
than sentence boundaries not coincident with such shifts. (Additional neuroimaging evidence
comes from (Zacks et al., 2001), who demonstrated transient changes in brain activity that were
time-locked to event boundaries during movie viewing.) In the present work, we conceptualized
ambiguous/mentalizing events as a type of small-scale narrative shift, at which time participants
are weighing new evidence and adjusting the “temperature” of their mental model of the story (i.e.,
the degree to which they believe there is something suspicious or nefarious going on).

However, we agree with the reviewer that some degree of evaluation and integration could also be happening online as participants listen to the event, and ideally the results from the regression would not depend on these methodological choices, which are somewhat arbitrary. To this end, we created a second version of the regressor, this time modeling the entire sentence as a mini-block by giving “1”s to all TRs in each of the labeled sentences. Fig. R1 gives a visualization of the difference between the original regressor (“Sentence offset only”) and this alternate version (“Whole sentence”):

Figure R1. The original and alternate versions of the “mentalizing events” regressor.

We then re-performed the regression analysis in the four ROIs using this new version of the regressor. Results are shown in Fig. R2, righthand column, with the original results shown in the lefthand column for ease of comparison (these are identical to the results depicted in Fig. 4 of the main text). Results were essentially unchanged; that is: (1) the left temporal pole and right medial PFC responded more strongly in high- relative to low-paranoia participants (Fig. R2b), and the effect held when paranoia was considered as a continuous variable (Fig. R2d); (2) the left posterior superior temporal sulcus (positive control) responded strongly in all participants, but there was no effect of paranoia (either categorical or continuous; Fig. R2b & f); and (3) there was no significant response in left Heschl’s gyrus (negative control), and there was no effect of paranoia, either categorical or continuous, on response magnitude (Fig. R2b & f). Thus we are confident that the results are robust to this methodological choice.

Mentalizing events: Sentence offset only

Mentalizing events: Whole sentence

Figure R2. Comparison of GLM results obtained using the sentence-offset (left) versus whole-sentence (right) regressors.

5. Why was the audio recording delivered in three consecutive runs, and not as a continuous
narrative (Pg 24, ln 555)? Did one of the runs included parts of the narrative that were more
ambiguous?

The audio recording was delivered in three runs (rather than one single long run) primarily to
allow us to periodically probe attention with multiple-choice comprehension questions, which we
delivered in between runs. To maintain the ecological validity of the paradigm and avoid biasing
participants' attention toward certain events, we wanted to avoid conducting such probes during
the stimulus itself. Using separate runs also allowed us to give subjects a small break, which
usually improves compliance and focus.

We did not deliberately structure the narrative to have more ambiguous events in the beginning,
middle, or end of the story. A post-hoc count indicated that there were 48 events that entered into
the mentalizing regressor, with 17, 13 and 18 occurring in parts 1, 2 and 3, respectively. We have
added this information to the Methods section of the revised manuscript (line 695).

6. The max ISC seems to be $r=0.16$ (Fig 2), which is relatively low. What was the threshold of
significant correlation ($r=?$)?

The reviewer is correct that the ISC values we observe are relatively low considering the full
range of possible r values $\{-1,1\}$. But because statistical significance (as determined by t -
statistic/ p value) is dissociable from effect size, it is possible to have a significant correlation
whose magnitude is somewhat low, as long as the effect is consistent across subjects/subject
pairs. In Fig. 2, we display raw ISC values for ease of interpretation, but the threshold above
which voxels are displayed is based not on raw correlation but on the t -statistic, which is
computed as $(z-z_{\text{null}})/S_z$, where z is the Fisher-transformed r value and S_z is the standard error of
z . (This can then be converted to a p value using the standard t distribution, and these p values
can in turn be converted to q values based on false discovery rate [FDR] correction, which is
how we arrived at the final visualization threshold for this figure.) So, two voxels with identical
287 t -statistics could have very different r values: for example, a voxel that has a mean ISC across all
288 subjects of $r = 0.02$ with very little variance around that mean, and a voxel whose mean
correlation is $r = 0.1$, but with high variance around that mean, might have very similar t -
statistics. Thus, because there is no direct mapping from t -statistic back to r value, it makes more
sense to discuss the threshold of significant correlation in terms of t -statistic. In the case of Fig. 2
(at FDR $q < 0.001$, which is quite stringent), the threshold of significant correlation was $t =$
4.097 .

There are a few reasons why the raw correlation values might be low. For one thing, these data
have high spatial resolution (voxel size 2.0mm^3), and smaller voxels have a lower signal-to-noise
ratio: for example, relative to a more standard voxel size of 3.0mm^3 , in 2mm isotropic voxels we
expect SNR to be reduced by a factor of ~ 3.4 (based on a change in volume from 27 to 8). Any
noise should not be temporally correlated across subjects, thus driving down r values. Another
effect of the smaller voxel size is that any imperfections in the registration to the standard
template, as well as normal anatomical and functional variation across brains, will add “noise” that
is really just differences in the spatial location of BOLD responses.

7. It is very difficult to read the labels in Figure 5a.

In the revised manuscript, we have attempted to improve the visualization of the feature loadings
in Fig. 5a.

Reviewer #2 (Remarks to the Author):

This is a very interesting report showing that neural and behavioral responses to the same
stimulus vary as a function of individual differences in trait paranoia.

The experimental paradigm used in this study is novel and ecologically valid.

The data collection and processing methods are well described and appear appropriate.

We thank the reviewer for his or her interest and positive assessment of our work.

My only questions/suggestions to these authors are:

1) How does your sample means etc relate to the norms for the Paranoid Thoughts Scale?

Our sample distribution on the Green et al. Paranoid Thoughts Scale, including mean/median, variance, and range, is similar to what has been observed using this scale in other samples of healthy volunteers. Note that this scale is designed to be used in both clinical and non-clinical populations, so higher scores are generally observed only in clinical populations, and the scores of healthy volunteers tend to follow an exponential (rather than Gaussian) distribution, which is also what we observe here. We note this information on line 129 of the manuscript.

2) Please justify your study design, i.e. creation of two groups by median split.

This was a concern that was shared by Reviewer 1. Please see our joint response to this point at the beginning of this response letter.

Reviewer #3 (Remarks to the Author):

This is quite rare for me, but I don't have a lot to say. I think this is a great paper. I found this work to be a compelling demonstration of the way that those high in paranoia process ambiguous social information differently than those low in paranoia and that this difference, at least in part, consists in greater mentalizing activity in the moments of ambiguity. I think more generally it adds to the literatures showing that high ISCs between individuals reflects more similar experience and processing of information as well as that various factors can prime us to experience things one way or another. To my knowledge this is the first study to use a personality variable as a chronic accessibility prime. The authors should probably look up Tory Higgins work on chronic accessibility since he made a similar argument decades ago (but without neuroimaging). Otherwise, though, I would strongly endorse this manuscript for publication.

We very much appreciate the reviewer's positive comments. We also thank the reviewer for bringing our attention to the work of Tory Higgins and the construct of chronic accessibility. It is indeed quite relevant to our work, and we have added a short discussion and citation in the paragraph beginning on line 382 of the revised manuscript.

REFERENCES

Cohen, J. (1983). The cost of dichotomization. *Applied Psychological Measurement*, 7(3), 249-253.

- Johnson-Laird, P. (1983). *Mental Models: Towards a Cognitive Science of Language, Inference*
*and Consciousness*. In: Cambridge, MA: Harvard University Press.
- Van Dijk, T. A., Kintsch, W., & Van Dijk, T. A. (1983). *Strategies of discourse comprehension*:
Academic Press New York.
- Whitney, C., Huber, W., Klann, J., Weis, S., Krach, S., & Kircher, T. (2009). Neural correlates
of narrative shifts during auditory story comprehension. *Neuroimage*, 47(1), 360-366.
- Zacks, J. M., Braver, T. S., Sheridan, M. A., Donaldson, D. I., Snyder, A. Z., Ollinger, J. M., . . .
Raichle, M. E. (2001). Human brain activity time-locked to perceptual event boundaries.
*Nature Neuroscience*, 4, 651. doi:10.1038/88486
- Zwaan, R. A., & Radvansky, G. A. (1998). Situation models in language comprehension and
memory. *Psychological Bulletin*, 123(2), 162-185. doi:10.1037/0033-2909.123.2.162

REVIEWERS' COMMENTS:

Reviewer #1 (Remarks to the Author):

The authors have adequately addressed my concerns, and I have no further comments. I recommend the paper for publication.

Reviewer #2 (Remarks to the Author):

I am happy with the authors' responses to my comments and the revised paper. I have no further comments.

Reviewer #3 (Remarks to the Author):

I recommended acceptance of the initial manuscript and continue to endorse this manuscript as high quality.